# A UAV Intelligent System for Greek Power Lines Monitoring

**DOI:** 10.3390/s23208441

**Published:** 2023-10-13

**Authors:** Aikaterini Tsellou, George Livanos, Dimitris Ramnalis, Vassilis Polychronos, Georgios Plokamakis, Michalis Zervakis, Konstantia Moirogiorgou

**Affiliations:** 1School of Electrical and Computer Engineering (ECE), Technical University of Crete, 73100 Chania, Greece; glivanos@tuc.gr (G.L.); mzervakis@tuc.gr (M.Z.); kmoirogiorgou@tuc.gr (K.M.); 2GeoSense, 57013 Thessaloniki, Greece; ramnalis@geosense.gr (D.R.); vpoly@geosense.gr (V.P.); 3Hellenic Electricity Distribution Network Operator S.A., 11743 Athens, Greece; g.plokamakis@deddie.gr

**Keywords:** power line inspection, deep neural networks (DNNs), UAVs remote sensing, RGB-thermal semantic segmentation

## Abstract

Power line inspection is one important task performed by electricity distribution network operators worldwide. It is part of the equipment maintenance for such companies and forms a crucial procedure since it can provide diagnostics and prognostics about the condition of the power line network. Furthermore, it helps with effective decision making in the case of fault detection. Nowadays, the inspection of power lines is performed either using human operators that scan the network on foot and search for obvious faults, or using unmanned aerial vehicles (UAVs) and/or helicopters equipped with camera sensors capable of recording videos of the power line network equipment, which are then inspected by human operators offline. In this study, we propose an autonomous, intelligent inspection system for power lines, which is equipped with camera sensors operating in the visual (Red–Green–Blue (RGB) imaging) and infrared (thermal imaging) spectrums, capable of providing real-time alerts about the condition of power lines. The very first step in power line monitoring is identifying and segmenting them from the background, which constitutes the principal goal of the presented study. The identification of power lines is accomplished through an innovative hybrid approach that combines RGB and thermal data-processing methods under a custom-made drone platform, providing an automated tool for in situ analyses not only in offline mode. In this direction, the human operator role is limited to the flight-planning and control operations of the UAV. The benefits of using such an intelligent UAV system are many, mostly related to the timely and accurate detection of possible faults, along with the side benefits of personnel safety and reduced operational costs.

## 1. Introduction

One of the main maintenance tasks of electricity distribution network operators is power line inspection, since power transmission networks sustain a wide coverage area and complex terrain, while they are heavily exposed to harsh natural environments, with the hidden risks of defects and line failures threatening the safety and stable operation of the power grid. This task is a crucial step for the early detection of faults prior to damage in the network. Moreover, it is necessary in the case of network damage to find the accurate location of the fault and take appropriate restoration actions fast. Regular inspections and timely maintenance involve on-the-ground staff and low-flying Unmanned Aerial Vehicles (UAVs) and/or helicopters. In many cases, power line inspections still take place by personnel on foot, which is very time-consuming, complicated (due to the high volume of equipment required to be transferred), and prone to human error during the visual data collection. The drawbacks of these kinds of inspection procedures include human safety hazards due to challenging terrain and weather conditions, and delay in the detection of faults in the case of missing power lines, since the inspection of the network by human observers is too slow, etc. On the other hand, forward-thinking grid operators adopt manned helicopters equipped with high-resolution cameras for data collection, which proves to be expensive and difficult to scale up. An end-to-end system that combines UAVs, optical sensors, and automated image data analysis using machine learning methods can cover each step of the inspection process in an accurate and robust way and may provide real-time alerts to relevant stakeholders for possible faults along with their exact location. The potential benefits of adopting drone technology as an attractive alternative for power line inspection also include reduced work time and labor costs, access to hard-to-reach areas, availability for more frequent monitoring, an improved overall carbon footprint, a reduced complexity, an increased reliability, platform portability, adaptability, and expandability through the incorporation of different sensors and data sources, along with focus on different segments of the power grid to detect multiple types of defects [1].

According to the “Drones in Energy Industry Report 2022” [2], the commercial drone market will globally reach USD 41.3B by 2026, with UAVs in the energy industry making up the biggest percentage of the corresponding market (estimated as approximately USD 6 billion), revealing the potential and challenges of this edge technology with direct applications in the inspection of oil, gas, electricity systems, and other critical infrastructures. The industry-wide shift towards renewable energy along with the direct need to monitor extended frameworks to link solar and wind parks to power grids is another ongoing challenge and potential of drone technology.

The use of UAVs equipped with camera Red Green Blue (RGB) and thermal sensors may help in developing an effective fault detection procedure. The independent operation of a UAV on pre-defined routes and the ability to analyze, in real time, the thermal and RGB optical data of the power lines in situ are challenging, since only few similar attempts are available. Our work focuses on the methodologies for analyzing optical data collected by HEDNO S.A. (Hellenic Electricity Distribution Network Operator S.A.) in both the Athens and Chania areas in Greece. The terrain inspected is quite diverse, with non-uniform wild vegetation covering the power lines across the video recording. The outputs of the presented methodologies are the structure and exact location of power lines, with the execution speed being relatively high so as to enable real-time, in situ processing and inspection procedures.

The primary objective of our study is to accurately locate electrical power lines. In the current status, we identify the absence of a power line as a fault. Furthermore, we exploit the meta-level information on the existence of three consecutive lines as an indicator of the normal power line concatenation. At this point, we emphasize that other types of faults on the lines, such as the existence of foreign objects or irregular wire formations, result in excessive heat generation or irregular temperature profiles, so the detection of such faults become feasible from the additional consideration of data/images from the thermal camera.

Recent quality review works [3,4,5] indicate the limitations, challenges, advances, trends, and prospects of the application of UAVs in the electrical industry and monitoring applications in general. Based on the conclusions extracted from the targeted study of this referenced literature, the contribution and novelty of our work can be summarized in the following:(a)The study proposes a joint approach of algorithms processing RGB and thermal video sequences detecting the presence of power lines and their exact geo-locations. This enables human operators to carry out repairs and maintenance work in a more timely and efficient matter, while decreasing safety hazards.(b)An improved version of our previous work [6] is presented, incorporating optical information from the infrared spectrum (apart from the visible one) and additional training from other real-scenario datasets in order to remove artifacts and outliers from the output images, leading to an even more robust and accurate line detection methodology.(c)A carefully designed methodology is adopted for drone-based data capturing through vigilant flight planning and vehicle navigation, taking into consideration the power line network surroundings and geo-location mapping of the pylons for executing missions under pre-loaded routes in the ground station, which is extremely important in mountainous areas, where high elevation differences between lines and wind corridors can complicate flights.(d)A custom-made drone architecture is developed fusing different kind of sensors and microcomputer edge technology for advanced in situ and on-board data processing. The developed prototype is among the very limited devices to combine both visual light and infrared cameras under a robust quadrotor vehicle-type to operate with an increased payload relatively close to the power grid, even under moderate wind speed conditions.(e)An adaptive and functionality expandable power infrastructure-monitoring UAV-based prototype is created, since the same hardware setup can be utilized for identifying different electrical components of the power network under a modified algorithmic scheme (i.e., the training of the deep neural model with different data and the utilization of temperature profiles from the thermal camera). In addition, the development and setup of the UAV platform are based on open-source software.(f)Benchmark unbiased datasets based on real data under different terrain and environmental conditions are created in both the visual and infrared spectra, providing a unique collection of registered and fully synchronized imagery that can be used to train and test machine learning algorithms and further improve their accuracy and efficiency. Limited open-access datasets fusing thermal and visual data, such as in [7], suffer from a low resolution of images and asynchronous, non-registered image samples for each scene.

The remainder of the manuscript is ordered as follows. Section 2 provides a short presentation of the related work on similar applications using either RGB or thermal data. The specifics of the proposed work are detailed in Section 3, while our results from two different case studies are presented in Section 4. Finally, the conclusion of this study is presented in Section 5.

## 2. Related Work

### 2.1. Detection of Power Lines Using RGB Data

Numerous attempts have been presented in research studies to apply deep learning algorithms for detecting power lines. The usual approach involves training models with a dataset that includes power line images and their corresponding binary masks. Following this, the pretrained model can generate a binary mask for any new images it encounters.

Most of studies with RGB UAV images utilize optimizations of the already available neural networks and/or image-processing methodologies. These optimizations include adding extra modules or modifying several parameters to achieve better results. Solilo et al. [8] utilized a color transformation framework for power line detection, fusing information from both the Hue Saturation Lightness (HSL) and Hue Saturation Value (HSV) color spaces, applying the Perspective Transform to UAV optical data to extract a bird’s eye view perspective binary image of the lines, and finally fitting polynomials between these lines. Although the proposed method efficiently estimated the position at which the power line changed direction relative to the UAV route and line tracking was achieved, it proved sensitive to brightness alterations. In addition, the validation procedure was performed on emulated data. Another interesting work is presented in [9] for autonomous power line detection using a drone framework based on open-source platforms. It compared a *Vision–LiDAR* and a *Dual-LiDAR*-based system, analyzing data in situ using on-board sensors. The system proved efficient for both line detection and autonomous navigation between pylons, yet the initial results provided were extracted within a new power line simulation validation environment before being brought into the physical UAV platform for outdoor testing. Current state-of-the-art UAV and 2D data-based power line inspection approaches were reviewed in [10], also addressing the potential of incorporating Lidar technology and its 3D information for supporting intelligent power line detection frameworks. However, the overall cost and complexity of such systems are increased. Diniz et al. [11] applied the YOLOv4 object detection model for developing an online drone navigation and path-planning framework, identifying power transmission lines and their relative positions to the UAV. Validation was performed on both synthetic and real-scenario data, aiming at recognizing or not any of the three key elements/classes of power transmission lines (simple circuit, double-circuit, or real circuit) and keeping the drone aligned with the power transmission lines under a relative constant height of about 35 m, for security reasons. However, this approach did not take fault diagnosis and alert notification into consideration. 

In their work, J. Gubbi et al. [12] used Histogram of Gradient features, rather than an actual image, to accurately capture line features. This approach yielded a robust F-score of 84.6%, outperforming the 81% achieved with the GoogleNet model. In another method [13], the final layer output and feature maps were combined to create high-level predictions. This involved using a convolutional neural network, integrating feature maps, extracting structured data from the coarsest feature map, and merging it all for a clear-background result. V. N. Nguyen et al. [14] introduced LS-Net, a fully convolutional design, which comprised three modules. This feed-forward architecture achieved a performance of 21.5 frames per second on a cutting-edge Graphics Processing Unit (GPU). L Yang et al. [15] proposed a novel vision-based power line detection network which used an embedded attention block to solve the problem of class imbalance and an attention fusion block for multi-scale feature fusion, improving the segmentation precision of power lines from aerial images. Another approach is that of G. Han et al. [16], who introduced G-UNets, a lightweight power line segmentation algorithm. This algorithm combined traditional convolution with a Ghost bottleneck in the encoder section and adopted a multi-scale input fusion strategy to minimize information loss. Furthermore, it incorporated Shuffle Attention (SA) in the decoding stage, aiming to boost the accuracy of the segmentation. To tackle the class imbalance issues in power line segmentation, Jaffari et al. [17] introduced a generalized focal loss function. The proposed loss function’s efficacy was evaluated using an enhanced U-Net model (ACU-Net), which included an additional convolutional auxiliary classifier head. Gao et al. [18] introduced an effective two-part network made up of a context branch and a spatial branch. The purpose of the context branch was to achieve more efficient, short-range feature extraction and provide a large receptive field, while the spatial branch was designed to keep detailed, high-resolution segmentation details. DUFormer, proposed by Deyu An et al. [19], is a specialized semantic segmentation algorithm utilized for detecting power lines in aerial imagery. This involves a process that starts with a token encoder for comprehensive feature extraction, subsequently employs a Transformer block for global modeling, and finally fuses the local and global features in the decode head to achieve the final segmentation result.

### 2.2. Detection of Power Lines Using Thermal Data

Thermal imaging enables the illustration of the otherwise invisible infrared spectrum, which covers wavelengths between visible light (the only part of the electromagnetic spectrum that human eye can “see”) and microwaves. An infrared (IR), or also referred to as thermal, camera works by detecting and measuring the infrared radiation emanating from objects (i.e., heat signature) and offers an easy, yet effective way of detecting temperature differences in industrial three-phase electrical circuits and networks, quickly spotting performance anomalies on the power transfer grid [20]. Combined with remotely controlled drone technology, it constitutes a great and fast tool for enhancing general recon capabilities in dangerous or difficult-to-observe conditions [21].

A detailed review of infrared thermography (IRT) for electrical energy infrastructure monitoring was presented in [22,23], describing theoretical aspects, state-of-the-art approaches, technical specifications, concerns, and challenges with respect to thermographic processes for electrical energy applications. It was emphasized that more robust and automated solutions for detecting the key components of electrical equipment need to be developed, since most of the existing approaches include manual segmentation based on trivial and widely established image-processing and computer vision methods. Another challenge lies in removing the noise and artifacts that usually impact the collected imagery in the field. Although several deep learning thermal-image-based defect detection approaches in electrical equipment and energy distribution networks have been developed, very limited ones focus on the recognition of power lines instead of other key components such as power transformers, circuit breakers, surge arresters, cutout switch bus fuse connections, and insulations [24,25,26,27]. He et al. [28] used the temperature information of infrared images to diagnose the fault of power transmission lines, applying the cellular automaton technique for separating the regions of interest and the background, the Hessian matrix for detecting the image transmission lines, and thresholding temperature information for deciding the power lines’ defects. The validation results indicated a true-positive rate of 93.56% and false-positive one of 2.38%, yet this algorithmic framework is not suitable for near-real-time and in situ assessments. A thermal-imaging-based convolutional neural network was described in [29] for fault diagnosis in high-voltage equipment, controlled by a threshold level based on outside conditions such as temperature and humidity. The method achieved an improved rate of detection compared to the reference once, However, the validation was performed on a limited dataset. A combination of RGB and infrared data was proposed in [30] to track power lines and detect faults and anomalies when analyzing UAV image data. The algorithmic scheme included edge detection approaches applied on intensity images, yet broken edges and artifacts introduced results in non-connected segments to represent powerlines. In addition, image registration from the two different input sources was not taken into consideration.

## 3. Proposed Methodology

In order to perform power line network inspection operations, several processes and tasks need to be completed. Our proposed framework combines artificial intelligence, different kinds of sensors, a custom-made UAV, and a data management platform to cover each step of the monitoring process, ensuring a harmonic and synchronized communication and information exchange through and between each stage.

To obtain the binary mask required for segmenting the power lines from their background, a combination of two distinct processing methods is employed. One method involves the processing of RGB data, while the other method focuses on the processing of thermal data. Initially, a binary mask is predicted for every power line image via a trained deep neural network. The thermal image is processed based on the Hough Transform for the detection of power lines, while the binary output obtained from this processing technique is employed as a complementary component to the RGB processing in order to improve the line detection accuracy. To achieve consistency of image information at the same time, both image registration and synchronized sensor triggering are required. The latter is performed through the drone navigation software of the ground control station, and the first is algorithmically achieved through image interpolation at a common image resolution (the “higher” one, that of the RGB camera) and image registration to ensure a commonly viewed image scene based on the a priori known sensor topology (distance of sensor centers in all three axes of the real-world space, flight height indicating distance from the ground) and characteristics (field-of-view and pixel size of each sensor). A more detailed description on the image registration procedure developed in the proposed framework is available in Appendix A.

The RGB binary output precisely delineates the lines of interest, but there are also artifacts present in the image, such as small regions or gaps between the lines. On the other hand, the segments extracted in the binary thermal image cover a wider area, resulting in thicker yet connected lines of an increased number compared to the actual power ones, under potentially various and multiple directions. This difference in thickness can be exploited to serve as a post-processing filtering mechanism for non-connected components and noise.

The fusion of RGB and thermal binary images is achieved by combining the binary masks resulting from the two procedures using a logical AND operator. Since only those areas where potential power lines are present in both the RGB and thermal images contribute to the final output, the pixel-wise logical AND operation helps in reducing artifacts in the background.

In addition, image morphology can be employed to deal with any slight gaps that might have occurred in the binary line structure during the entire segmentation process. More specifically, the two basic mathematical operations, opening and closing (under a five pixel-sized and square-shaped structuring element parameter setup), are utilized in sequential mode to first remove small noise fragments and finally bridge minor gaps in the structure, while maintaining the form and shape of the line.

The synthesis of RGB and thermal images, enhanced by the application of morphological operators, enables the effective and accurate identification of power lines, by combining the advantages of each segmentation method and imaging modality while overcoming their limitations. In Figure 1, a flowchart of the proposed method is presented.

The proposed algorithmic framework focuses on the detection of power lines and not of other types of electrical equipment, which is part of the future improvement of the presented study and will be attempted using other image segmentation approaches. Towards this direction, the flight plan along with the camera topology are specifically designed for capturing the image data of the proper setup to sustain specific requirements:The drone is flying over the power network at a relatively close distance (~15–20 m above the ground), so that power lines are clearly visible and positioned as close as possible to the center of the captured image.The drone speed is relatively slow to facilitate time-efficient video processing across the entire route without “empty” and “unprocessed” segments of the power network, while pillar Global Positioning System (GPS) coordinates are loaded in the flight mission plan to enable the smooth navigation of the vehicle.The camera sensors are positioned vertically with respect to the ground under a gimbal topology.

### 3.1. RGB Data Processing

#### 3.1.1. Architecture

The D-LinkNet Architecture is the most appealing method for the purpose of this work, since it demonstrates an excellent performance on image segmentation tasks, especially when it comes to linear structures. D-LinkNet consists of three main parts, an encoder, center part, and decoder, as can be seen in Figure 2. The major advantage of this network is the dilated convolutional layers in its center part. ResNet34 [31], which is pretrained on ImageNet [32], is used as the encoding part. The decoder part remains the same as in the LinkNet architecture [33]. The center part contains dilated convolution, both in cascade mode and parallel mode. The dilation rates of the dilated convolution layers are 1, 2, 4, 8, and 16.

#### 3.1.2. Datasets, Equipment, and Data-Capturing Framework

In preparation for an inspection and data-capturing flight, available Geographic Information System (GIS) data on pylons and terrain mapping are loaded into the UAV ground station to create an optimal drone route path over the area of interest at an approximate height of 5 m above the power transmission lines. In addition, the designed flight plan is submitted to the Civil Aviation Service for approval, keeping with all the safety rules and regulations.

The proposed custom-made UAV platform is based οn a quadcopter framework, capable of carrying increased loads and providing the necessary functionality for the needs of the proposed application, such as an increased autonomy, expandability, proper power supply for microeletronics, smooth navigation, and motor management under low speeds. In addition, it meets all the requirements for operating in accordance with the applicable regulatory framework and can be adjusted to a variety of big infrastructure inspection missions under varying weather conditions. The main processing core of the drone is an NVidia GPU-enabled microcomputer, providing an increased time efficiency towards the in situ data processing and proper handling of the attached sensors’ information. The flight mission is separately manipulated by the own micro-computer control unit of the aerial vehicle, which communicates with the ground control station. The optical system consists of a high-resolution RGB camera and a thermal one attached to the drone skeleton under a gimbal topology and fixed/known distances (this enables the registration of the captured imagery), assuring that both image sensors are always placed in parallel with each other and vertically with respect to the ground, independent of the drone flight angle and movement pattern. To ensure video synchronization, the two cameras are simultaneously triggered by the ground station software. The technical specifications of the a onboard processing units and camera sensors are summarized in Table 1. The key components of the UAV platform along with the connection topology are illustrated in Figure 3.

Our custom-made vehicle prototype is a long-range and long-endurance quadcopter. It uses four 6S 22,000 mAh lithium polymer semisolid-state batteries in a 2 in series first and then the two 12S sets are in parallel, giving a total of a 12S (50 V) and 44,000 mAh capacity. With the existing payload of the dual cameras, synchronized camera triggers, four in total processing units, video convertors, and a number of DC-to-DC Power Supply Units (PSUs) for all the processing units, the UAV-platform achieves an average consumption of 40–45 amp, which gives it a rough flight time of 50 min and even a little more under a coverage distance of nearly 10 km with a low speed of 2.5 m/s.

The Ground Control Station (GCS) uses 2.4 GHz Industrial, Scientific, and Medical (ISM) communication with the UAV, from which there is actual remote control of the vehicle, a Mavlink stream for telemetry data, and a live high-definition (HD) video stream. The image resolution is cable @720p @30 fps|1080p @30/60 fps and currently uses 1080 at 30 frames. In addition, the Handheld GCS Bluetooth/WIFI/GPS module is used to stream the Mavlink to Mission Planner 1.3.84, a Laptop GCS software, for the operators’ spotter to also receive detailed information on the UAV performance.

The UAV outputs pulse-width modulation (PWM) and Mavlink commands for camera control. Due to the fact that the FLIR thermal sensor is PWM-activated and the SONY visual light one is capable of being activated over its Multiport using Precision Time Protocol (PTP), we use a single PWM signal for both triggers and tune the PTP trigger in order for both cameras to start simultaneously. From all the testing so far, for the moment, a trigger command is given from the handheld GCS, and the average reaction time is estimated at 300 ms. It is noteworthy that a completely non-commercial custom cable is built, as is the corresponding firmware for the connectivity of the RGB image sensor to the main drone computer board, in order to achieve the dual-camera functionality and synchronization.

Το achieve good detection results in diverse terrain, we choose to train the D-LinkNet model on datasets that contain power lines in both mountain and urban scenes. The datasets, named “Power Line Dataset of Mountain Scene” (PLDM) and “Power Line Dataset of Urban Scene” (PLDU), were initially introduced in [10] and are available online for free. To further enhance the training process, additional data are used, resulting in a dataset of 771 training samples and 185 testing samples in total, all presenting a “healthy” power network of 3 lines. These data are provided by the HEDNO S.A. (Hellenic Electricity Distribution Network Operator S.A.) department that administers the network in the Chania area, Crete Island, Greece. The proposed algorithmic scheme focuses on the accurate segmentation of power lines, however, an easy-to-use fault detection rule is incorporated into the monitoring procedure to produce alarm messages when only two or one power lines are detected.

As an additional improvement of the training stage, data augmentation strategies are utilized to artificially expand the data availability. Data augmentation is performed dynamically, creating varied versions of each image in the dataset during each epoch of training. Specifically, during each epoch of the training process, every individual image is subjected to a series of augmentation techniques, resulting in an augmented dataset that is 771 times larger than the original dataset. The techniques used include random rotation, flipping, zoom, contrast, and brightness adjustment. Table 2 provides detailed information about both the training and testing datasets.

Enhancing the dataset with the use of a data augmentation technique substantially increases the diversity within the training data, which enables the model to identify lines in a broader range of conditions, significantly improving its performance on unseen data and mitigating the risk of overfitting.

#### 3.1.3. Segmentation Process

We propose segmenting the power line images using a grid approach. The DLink-Net divides the input image of size into a grid and predicts an output mask for each grid cell separately. In the current study, we test a 4 × 5 grid approach. As the grid size decreases, the thickness of the detected power lines increases. Using a bigger grid, the detected line is thinner and more precise to its actual size. The grid size can be adjusted based on the distance between the UAV camera and the power lines that need to be identified.

#### 3.1.4. Implementation

The network was implemented in Python using Pytorch, on an NVIDIA GeForce GTX 1650 Ti GPU. A learning rate of 0.001 was set, an optimal value for achieving steady training progress. To further fine-tune the model, the Adam optimizer was employed. Binary Cross-Entropy (BCE) loss function was used to gauge the error between the prediction output and the provided target value. The model was trained for six epochs, which was enough for it to accurately recognize the structure of the power lines. Furthermore, ResNet18 was adopted as the encoder in D-LinkNet.

### 3.2. Thermal Data Processing

An analysis of the thermal images for extracting power lines is performed, applying Probabilistic Hough Transform [35,36], a widely established yet effective and robust approach for detecting known shapes that can be represented through mathematical formulas. Compared to the initial Hough Transform, the Probabilistic one constitutes an improved and advanced version of it, capable of identifying both the start and end points of line segments and allowing for a more accurate detection of complex shapes in images and continuous line following, connecting gaps and holes between extracted segments. In our proposed implementation scheme, the Transform is applied to a thresholded output of the V component of the Hue Saturation Value (HSV)-transformed instance of a thermal drone-captured image sequence. The parameter setup of the algorithm is as follows: (a) distance resolution of the accumulator = 1 pixel, (b) accumulator threshold parameter = 50, meaning that only those lines that get enough votes are returned (larger than the threshold), (c) angle resolution of the accumulator = π/180 radians, (d) minimum line length = 200 pixels, meaning that line segments shorter than this value are rejected, and (e) maximum allowed gap between points on the same line to link them = 10 pixels. The resulting image of this algorithmic stage represents the ideal and long line segments present in the scene and serves as an indicative guide for smoothing the contours, connecting the gaps, and removing the outliers present in the RGB deep-learning-based extracted sample of the previous processing step.

## 4. Results

At first, the proposed method was evaluated on multiple frames extracted from the UAV-captured videos, provided to us by the HEDNO S.A. department. To measure the performance of the approach, 35 video frames were annotated using the LabelMe annotation tool [37], which is available online for free. All the annotated images were converted into binary ground truth masks. A sample frame of the videos acquired by HEDNO S.A. is shown in Figure 4, along with its corresponding generated ground truth mask.

Table 3 presents metrics, such as Accuracy, Precision, Recall, F1-Score, and Specificity, which are calculated using Equations (1)–(5), respectively, to demonstrate the effectiveness of the method after validation on four different real-scenario datasets containing 35 images under different terrain and lighting conditions.
Accuracy = (TN + TP)/(TN + FP + TP + FN)(1)
Precision = TP/(TP + FP)(2)
Recall = TP/(TP + FN)(3)
F1-Score = 2 (Recall · Precision)/(Recall + Precision)(4)
Specificity = TN/(TN + FP)(5)

The accuracy measure evaluates the overall efficiency of the model’s predictions by considering both positive and negative samples. The F1-score is an index used to measure the predictive performance of a model. It combines precision and recall, which are two otherwise competing metrics. Specificity measures the model’s ability to correctly identify negative samples out of all the actual negative samples. It focuses on minimizing false positives. The TP, FP, and FN stand for true positive, false positive, and false negative, respectively. These values are calculated using confusion matrices.

The results in Table 3 show that the proposed method combining both RGB and thermal processing outperforms the single-modality RGB processing utilizing the trained D-LinkNet for binary mask generation. The higher accuracy indicates that the combined processing provides an higher overall correctness in its predictions. The lower precision of the sole D-LinkNet model suggests that it is more likely to produce false positives, which is indicative of artifacts or noise present in its predictions. The presence of artifacts and noise in the predicted binary masks is evident upon a visual inspection of Figure 5, which displays the generated outputs from both D-LinkNet and D-LinkNet combined with Thermal Processing. Examples of “clearly” detected power lines using our proposed methodology are depicted in Figure 6. Apart from detecting power lines, the overall implementation framework measures the number of parallel lines detected in the scene and, in the case of missing one(s), produces an alert/notification for fault presence due to cable damage. By matching the timestamp of each defect detection event through an online/offline video analysis with the log file of the flight (containing, among others, the GPS data recordings), the exact geolocation of the “faulty” power lines network segment is recovered, which is crucial for proper and immediate actions by stakeholders. It is important to mention that the average processing time for each frame under the programming code deployment on the GPU-enabled board and preliminary testing is about 2.3 s, revealing the potential of this proposed power line inspection methodology for in situ and/or on-board assessments.

## 5. Conclusions

The combination of two different data-processing methods applied to the data obtained from UAVs appeared to improve fault detection results. Specifically, these two processing methods act in a complementary manner, where one method (RGB Processing) accurately identified the lines in detail, while the other method successfully eliminated incorrect artifacts of the processing. Through the proposed custom-made UAV platform and integrated optical data analysis framework, a robust, accurate, cost-effective (both in terms of intelligent drone platform development and service provision costs for end-users under a long-scale and long-term utilization basis), and adaptive tool for power lines inspection was developed and validated, revealing its potential for automated assessments in the field. Regarding the time consumption, preliminary tests on the execution time of the initial version of the python code developed for the proposed power-lines-monitoring framework revealed an average time of 100 s per frame on a CPU (Intel(R) Core(TM) i5-8300H CPU @ 2.30 GHz), 14 s per frame on a Field Programmable Gate Array (FPGA—KV260 Xilinx model) module, and 2.3 s per frame on a proposed GPU-enabled microcomputer system (Nvidia Jetson AGX Xavier, Nvidia, Santa Clara, CA, USA), revealing the potential of our system for in situ data analysis applications. Our study based on drone inspection and fused optical data analysis built and provided a dataset of historical, unbiased imagery records, facilitating the identification of critical areas and enabling a study on power grid status alterations over time and the quick dispatch of service teams upon fault event detection. The challenges of applying and adopting this technology include: (a) actions for involving licensed and trained personnel who officially know Federal Aviation Administration (FAA) drone regulations, (b) intensive electrical safety training when dealing with high-voltage electrical networks, (c) the proper calibration and usage of camera sensors to capture quality imagery data, and (d) adaptation to the new era coming soon when flight regulations are relaxed and allow drones to cover bigger areas and eventually operate largely autonomously, leading to Beyond Visual Line Of Sight (BVLOS) flights under limited human intervention and at low altitudes over pipelines and power lines. The future improvement of our work includes decreasing the processing time through parallelization and code optimization, thermal image processing based on temperature profiles extracted through properly selected raw image formats, and fault detection on additional components of power transmission networks other than power lines.

## Figures and Tables

**Figure 1 sensors-23-08441-f001:**
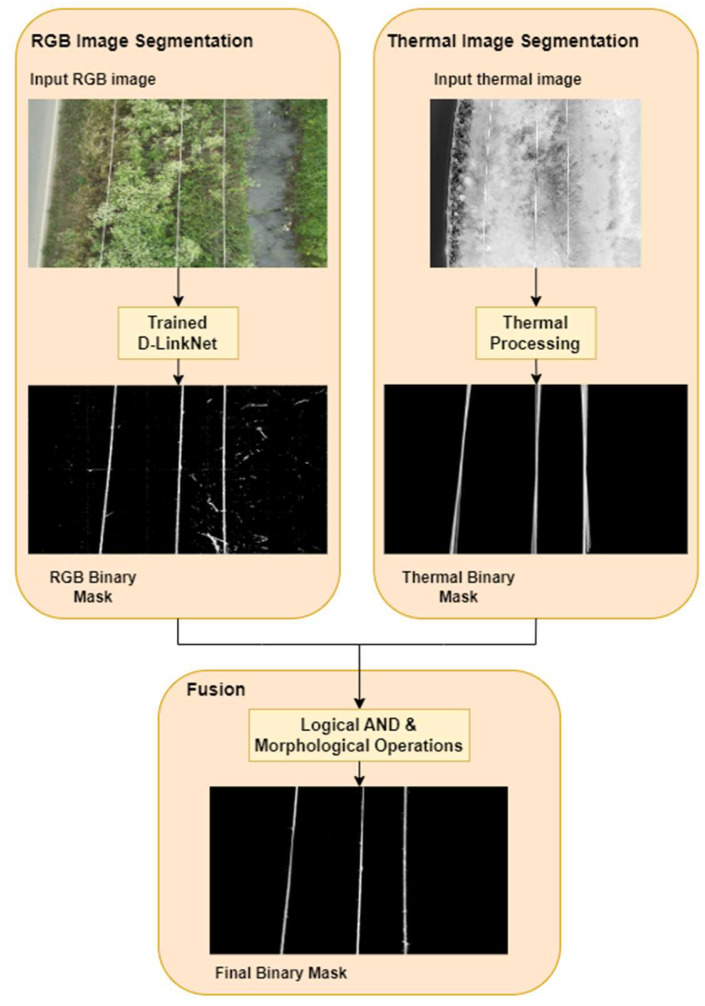
Flowchart of proposed segmentation methodology.

**Figure 2 sensors-23-08441-f002:**
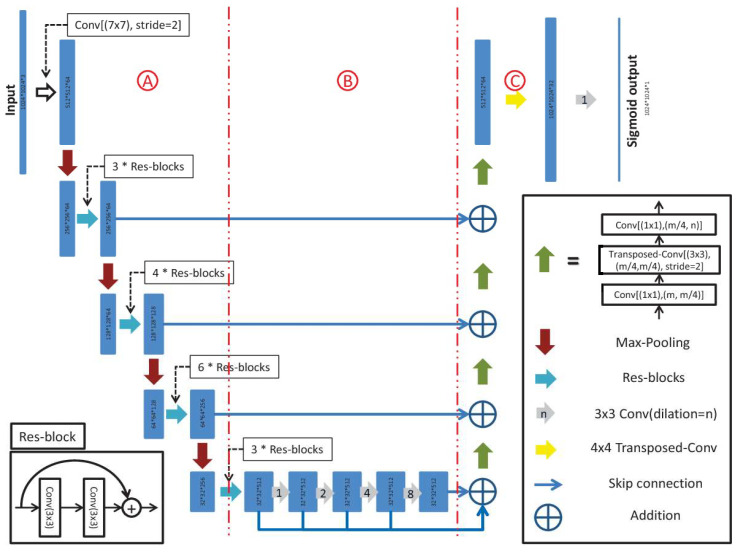
D-LinkNet Architecture [34].

**Figure 3 sensors-23-08441-f003:**
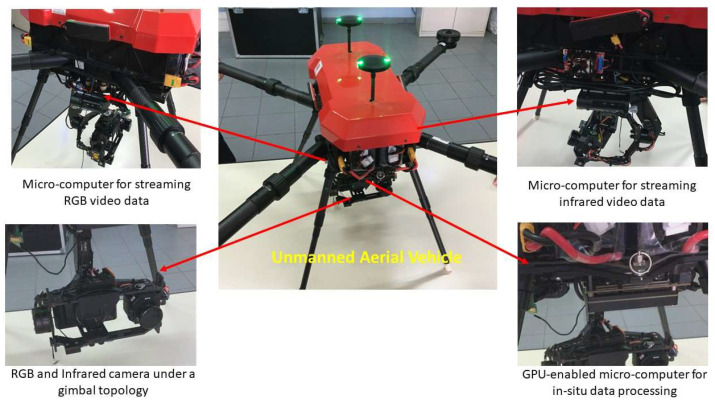
Proposed UAV-based power line inspection platform and key components.

**Figure 4 sensors-23-08441-f004:**
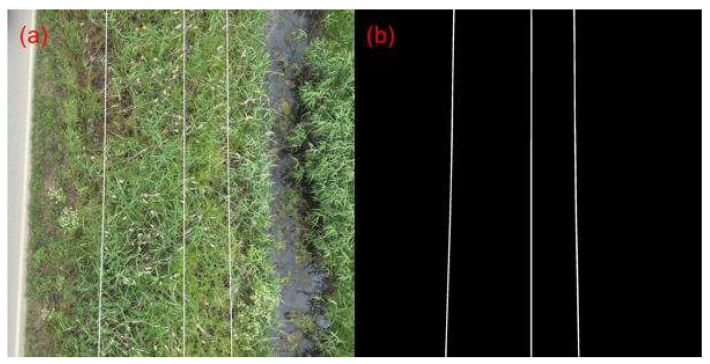
HEDNO S.A. dataset sample, (**a**) video frame, and (**b**) corresponding binary mask.

**Figure 5 sensors-23-08441-f005:**
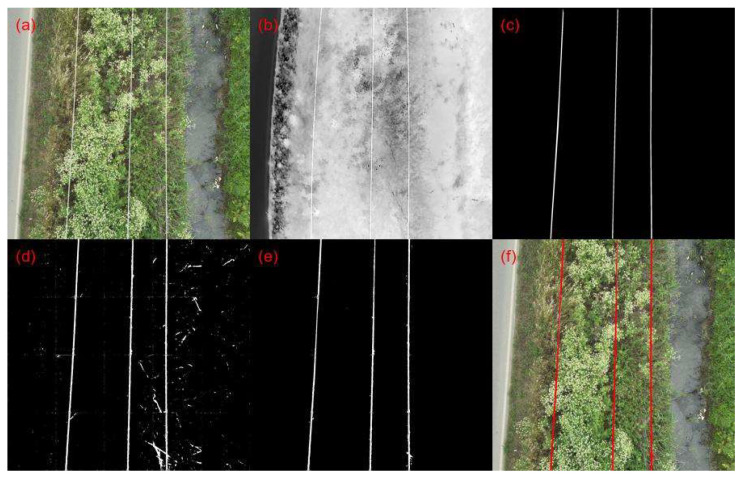
Power line detection results on a sample video frame. From (**a**–**f**): (**a**) RGB Video Frame, (**b**) thermal video frame, (**c**) ground truth image (**d**) D-LinkNet output mask, (**e**) proposed method output mask, and (**f**) final segmented frame.

**Figure 6 sensors-23-08441-f006:**
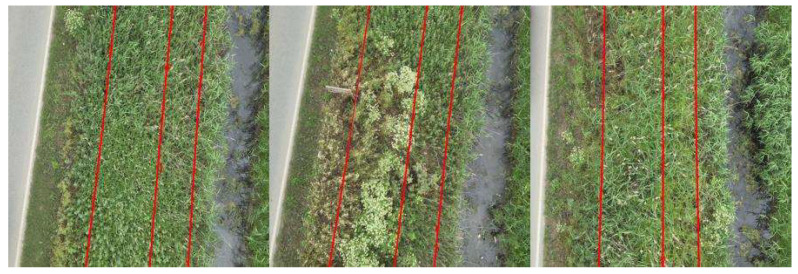
Examples of detected power lines using proposed methodology.

**Table 1 sensors-23-08441-t001:** Technical characteristics of proposed custom-made UAV-based power lines inspection platform sensor load.

Component	Model	Specifications/Functionality
On-board processing unit	NVIDIA Jetson AGX Xavier	Micro-computer system for in situ data processing and control of video streaming to ground station for drone manipulation GPU enabled for increased Artificial Intelligence performance and reduced processing time
RGB camera	Sony Cyber-shot DSC-HX90V	Small-sized travel zoom category camera 1080 p, 30 fps selected functionality mode
Thermal camera	FLIR Vue Pro	Lightweight compact infrared imaging camera for precise thermographic and radiometric aerial imaging Connectivity to MAVlink compatible autopilot and/or R/C PWM outputs 640 × 512 resolution, 30 fps selected functionality mode
Additional units	Raspberry PI 4 Model 4 GB	Intermediate connectors between cameras and main data-processing unit Two units attached to drone skeleton, one for each camera Enable data streaming to multiple sources under different connectivity ports/modules

**Table 2 sensors-23-08441-t002:** Dataset details.

Dataset	Training Samples	Testing Samples
PLDU	453	120
PLDM	237	50
HEDNO S.A. Video Frames	81	15
Final Dataset	771	185
Augmented Dataset	Epoch Number * 771	185

**Table 3 sensors-23-08441-t003:** Model performance metrics.

Architecture	Accuracy	Precision	Recall	F1-Score	Specificity
D-LinkNet	0.97	0.67	0.98	0.75	0.97
D-LinkNet + Thermal Processing	0.99	0.80	0.96	0.86	0.99

## Data Availability

Data is unavailable due to privacy restrictions.

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
