# Peer review of "A UAV Intelligent System for Greek Power Lines Monitoring"

_sensors, 2023, doi:10.3390/s23208441_

Round 1

Reviewer 1 Report

This article is of some value. Before it can be processed further, some issues need to be processed, as follows.

1. In the Abstract section, the main method of your innovation or improvement have not been introduced, include the details.

2. As the feature or advantages of this paper is that it adopts the combination of RGB data and thermal data, why cannot you introduce the method in details in Section “3. Methodology”? Taking Figure 1 for example, how the “Image filtering logical and Morphological Operations” performed? What are the key parameters?

3. When you use RGB data and thermal data at the same time. How do you decide the same area form two different sensors? Do you need register the two sensors?

4. In section Conclusion. “Through proposed custommade UAV platform and integrated optical data analysis framework, a robust, accurate, cost-effective and adaptive tool for power lines inspection”. How to embody cost-effective? I have not seen any experiment about the time consumption.

5. Other issues or wrong expressions.

(1) What isspeedss”in line 253and graound-control station”?

(2) Many symbols are not right. For example, in “3.1.3 Segmentation Process:”, Colons is redundant. A comma is needed in “In Figure 1 a flowchart of the proposed method is presented”. etc.

I thins some English expressions are not good.  Because some words are not easy to digest. 

Author Response

Multiple correspondence issue was resolved.

We would like to thank the reviewers for the insightful evaluation of our manuscript (Manuscript ID: sensors-2540685) and their valuable comments. We have carefully considered the amendments proposed and incorporated the necessary changes to address the suggestions provided by the reviewers, hoping that the paper quality has been improved and meets the journal requirements.

Below we provide the point-by-point response to the reviewers’ comments. All modifications in the manuscript have been highlighted in yellow color.  

Reviewer 2 Report

1.       What is the difference of requirements for the algorithm to detect the power line and other electrical equipment? For example, transformer. How to meet the requirements in the paper?

2.       How are the datasets composed of? Are there any samples presenting faults on the power lines?

3.       How does the thermal pictures’ temperature index contribute to the detection?

4.       The paper provides detection of abnormal power line state neither from contour, nor from temperature. How is the function “power line monitoring” realized?

Author Response

We would like to thank the reviewers for the insightful evaluation of our manuscript (Manuscript ID: sensors-2540685) and their valuable comments. We have carefully considered the amendments proposed and incorporated the necessary changes to address the suggestions provided by the reviewers, hoping that the paper quality has been improved and meets the journal requirements.

Below we provide the point-by-point response to the reviewers’ comments. All modifications in the manuscript have been highlighted in yellow color.  

Reviewer 3 Report

The paper presents an UAV based smart power line inspection system, using UAV mounted camera sensors operating in visual and infrared  spectra. The paper presents an overall fault detection methodology with elements of novelty in the fusion of multichannel imaging data and the huge technical effort on real data. The main limitation of the paper is in the presentation of the contribution, and I suggest a few major revisions.

1.References

Overall, the authors could describe the novelty of the approach in more detail; in particular, they should cite the recent review in [1] that provides interesting summarization of recent contributions, and can be used to specify the novelty of the proposed approach.

2.System description

It seems that introducing the problem the paper lacks of a system description section; the authors should introduce a figure with the architecture: which is the typical coverage area? drone speed? Duration of the flight?  

It seems that the UAVs stream the video in real time. The authors should provide details of the video streaming, e.g. rate , network support (cellular network connection?), is the performance related to the network connection capacity? Which are the actual settings for the experiments?

The visible light and infrared cameras are synchronized; highlight the signaling for synch and driving purposes, providing details: which is the radio technology? Any delays?

3. Methodology description

A huge literature on image segmentation, and  pattern detection exists, and the use of multichannel images has been widely studied; it seems that here the novelty is in the overall technical solution rather than in algorithm details.

To highlight the novelty, improve the description of the overall detection scheme (segmentation, fusion, detection) in the dedicated figure; for the sake of clarity, variable names should be given to the data at the input and output of the blocks of the scheme.  The basic notation should be provided in the text. Besides, more details should be provided on the thermal processing and morphological filtering, using rigorous analytical description of the different stages, with the goal of making the methodology reproducible to the readers.

4. Results

The detection performance refer to the case of unbalanced classes of data (line faults are relatively unlikely). For such data recall and accuracy may not be the most  suited tools for performance evaluation. Further results (e.g. using AUC ROC) should be provided.

Cite the source of the dataset PLDM and PLDU.

Explain better what the last row of Table 2 (“Epoch Number * 771”) means.

5. Minor revisions

Check item letters in sec.1.

[1] Foudeh, H.A., Luk, P.C.K. and Whidborne, J.F., 2021. An advanced unmanned aerial vehicle (UAV) approach via learning-based control for overhead power line monitoring: A comprehensive review. IEEE Access9, pp.130410-130433.

Author Response

(The authors gave the same response as above.)

Round 2

Reviewer 1 Report

The authors have addressed all my concerns. I would like to accept it.

Minor issues:

It would be more suitable to be lower-case for the full name of the specialized terminology, such as  Unmanned Aerial Vehicles (UAVs) should be unmanned aerial vehicles (UAVs), etc. 

Author Response

We would like to thank the reviewers for the insightful re-evaluation of our manuscript (Manuscript ID: sensors-2540685) and their valuable comments. We have carefully considered the amendments proposed and incorporated the necessary changes to address the suggestions provided by the reviewers, hoping that the paper will be finalized for publication.

Below we provide the point-by-point response to the reviewers’ comments. All modifications in the manuscript have been highlighted in yellow color.  

Reviewer 2 Report

1.     The authors did not response to the first comment directly. Is there any algorithm special for the power line only in the article?

2.     The database in the article should include samples of fault line for training.

3.     If detection of power lines in presented study does not consider temperature assessments, an innovative hybrid approach that combines RGB and thermal data processing methods is not the contribution of this article.

4.     There is not “section 3.2.1” in the article.

5.     The article presents more contents of the platform with the products than the faults detection algorithm.

Author Response

(The authors gave the same response as above.)

Reviewer 3 Report

The paper Is improved, It includes relevant information, my comments have been addressed. I still encourage the authors to introduce a system level figure outlining the uavs, the ground antennas, the signaling and so on.

Author Response

(The authors gave the same response as above.)

Round 3

Reviewer 2 Report

In the present manuscript, a fault implies non-existence of such a linear structure. It would be better if more types of fault are included, such as foreign objects.

Author Response

(The authors gave the same response as above.)
